# Fungal Contamination of Building Materials and the Aerosolization of Particles and Toxins in Indoor Air and Their Associated Risks to Health: A Review

**DOI:** 10.3390/toxins15030175

**Published:** 2023-02-25

**Authors:** Mohamad Al Hallak, Thomas Verdier, Alexandra Bertron, Christine Roques, Jean-Denis Bailly

**Affiliations:** 1Laboratoire Matériaux et Durabilité des Constructions (LMDC), INSA Toulouse, 135 Avenue de Rangueil, 31400 Toulouse, France; 2Laboratoire Génie Chimique (LGC), Université de Toulouse, CNRS, 35 Chemin des Maraîchers, 31400 Toulouse, France; 3École Nationale Vétérinaire de Toulouse, 23 Chemin des Capelles, 31076 Toulouse, France; 4Laboratoire de Chimie Agro-industrielle (LCA), Université de Toulouse, INRAE, INPT, 4 Allées Emile Monso, 31030 Toulouse, France

**Keywords:** indoor air quality (IAQ), bioaerosols, fungi, building materials, aerosolization, mycotoxins

## Abstract

It is now well established that biological pollution is a major cause of the degradation of indoor air quality. It has been shown that microbial communities from the outdoors may significantly impact the communities detected indoors. One can reasonably assume that the fungal contamination of the surfaces of building materials and their release into indoor air may also significantly impact indoor air quality. Fungi are well known as common contaminants of the indoor environment with the ability to grow on many types of building materials and to subsequently release biological particles into the indoor air. The aerosolization of allergenic compounds or mycotoxins borne by fungal particles or vehiculated by dust may have a direct impact on the occupant’s health. However, to date, very few studies have investigated such an impact. The present paper reviewed the available data on indoor fungal contamination in different types of buildings with the aim of highlighting the direct connections between the growth on indoor building materials and the degradation of indoor air quality through the aerosolization of mycotoxins. Some studies showed that average airborne fungal spore concentrations were higher in buildings where mould was a contaminant than in normal buildings and that there was a strong association between fungal contamination and health problems for occupants. In addition, the most frequent fungal species on surfaces are also those most commonly identified in indoor air, regardless the geographical location in Europe or the USA. Some fungal species contaminating the indoors may be dangerous for human health as they produce mycotoxins. These contaminants, when aerosolized with fungal particles, can be inhaled and may endanger human health. However, it appears that more work is needed to characterize the direct impact of surface contamination on the airborne fungal particle concentration. In addition, fungal species growing in buildings and their known mycotoxins are different from those contaminating foods. This is why further in situ studies to identify fungal contaminants at the species level and to quantify their average concentration on both surfaces and in the air are needed to be better predict health risks due to mycotoxin aerosolization.

## 1. Introduction: Importance of Moulds as Indoor Contaminants

The biological contaminants of the indoor environments include fungi, bacteria, viruses, pollen, etc. [1]. However, the ability of fungi to grow on almost all building materials, whether natural or synthetic, especially if they are hygroscopic or wet [2], necessitate the study of their development on such substrates. Water activity (amount of free water available for microbial metabolism) is considered the most impacting factor for fungal development [3]. Researches have emphasized that many kinds of materials are susceptible to growth once there is a sufficient amount of available water: wood, gypsum boards, wallpapers, mortars, etc. [4]. In parallel, according to various intrinsic parameters, including chemical composition, pH, presence of dust, etc., building materials can present different bioreceptivities, i.e., susceptibilities, to support and favour mould growth [5,6,7,8]. Moreover, their development is governed by various environmental factors, including temperature and relative humidity, and their spread in the indoor environment can also involve other factors, including air exchange rate, air movement, building structure and location, design and ventilation system; however, this list is not exhaustive [2].

Due to their reliance to water activity, water-damaged buildings are highly sensitive sites in terms of indoor fungal development [5]. However, the increase in water content on/into building materials may also be caused by plumbing [9] and water vapour condensation on the walls in strongly insulated buildings [10]. In Northern Europe and North America, the first works on indoor contamination by fungi were published in the 1970s. This problem increased strongly with time in conjunction with changes in human activities and the increasing time spent inside buildings (more than 80% of the time in industrialized countries). Studies showed that 20% to 40% of buildings in these regions display a visible mould presence [11,12]. Moreover, various building types were reported to present moisture problems and subsequent fungal contaminations, including homes, schools, workplaces and hospitals, representing many sources of exposure to toxic compounds [13,14,15]. Indeed, in parallel with their growth, moulds are able to release biological particles (spores, mycelium fragments, etc.) and toxic compounds (allergic molecules, mycotoxins) into the air, referred as bioaerosols. 

Bioaerosols correspond to aerosols involving microorganisms, such as fungi, bacteria and viruses, or organic compounds emanating from microorganisms, such as endotoxins, metabolites, toxins, etc. [16,17]. The biological part of bioaerosols forms approximately 50% of all aerosol particles [18], and their sizes usually range from 0.001 nm to 100 μm [19]. They are released from surfaces into the indoor air either by active processes that are specific to the moulds’ forms of development or by passive processes that mostly rely on air movement in buildings due to human activity or ventilation. These aerosolization processes contribute to the degradation of indoor air quality (IAQ) and may subsequently affect the health of occupants due to the inhalation of toxic particles, leading to allergies or respiratory troubles [16,17,19,20,21]. 

Indeed, depending on their size, particles can be inhaled by exposed individuals and infiltrate different parts of the lungs (from the trachea to the bronchioles), leading to various health problems [22,23]. Even if the direct causality between the presence of moulds and specific diseases among exposed individuals is difficult to demonstrate, strong associations have been reported in many works [24,25,26,27] and such a direct relationship is also highly suspected by health authorities [27]. In addition, it should be emphasized that different fungal genera and, more precisely, different fungal species pose different health risks for humans. For example, some species belonging to the *Aspergillus* genera, such as *Aspergillus flavus* [28,29,30,31] or *Aspergillus fumigatus* [32,33,34], are well known to play specific role in three different clinical settings in humans: opportunistic infections, allergic states and toxicosis [34]. *Cladosporium* species are rarely directly pathogenic to humans, even if they have been sometimes reported to cause infections of the skin and lungs. However, the spores of *Cladosporium* species are significant allergens and their presence in large amounts in the air can severely affect people with asthma and other respiratory diseases [35,36]. By contrast, *Penicillium* species are diverse and widely distributed in the environment, but despite their abundance and diversity, they are not commonly associated with human and animal infections [37]. 

The identification of fungal species present in buildings is therefore quite important, as it also provides information on the types of biological particles that are likely to be released into the air together with the corresponding health threat they may represent. 

It is reasonable to assume that fungal populations from indoor surfaces may have an impact on indoor air quality due to the aerosolization of different particles. In France, health institutions have underlined the importance of both aerial and surface sampling to identify indoor mould contamination and evaluate subsequent health consequences [28,38,39]. In addition, it is also known that fungal populations from the outdoors, including toxigenic species, may also impact the fungal diversity that is identified indoors [40,41,42]. 

In this paper, we reviewed the literature available on the mould contamination of indoor materials and indoor air in different types of buildings. The aim was to highlight the links, if any, between microbial growth on indoor building materials and subsequent air contamination and the resultant risk for human health. 

The first part of this paper introduces the aerosolization processes of microbial particles from the surface of materials. The mechanisms of aerosolization and the different factors impacting the release of particles from contaminated surfaces into the indoor air are presented. The second part is an overview of data on fungal contamination limited to the accessible surface of building materials. In different buildings types, most identified fungal species are presented along with the sampling and analysis methods carried out by authors. Then, the third part focuses on the links between mould contamination on surfaces and air quality. The quantification of airborne species is discussed. Finally, the last part reports the toxic effects of mostly identified species by presenting the mycotoxins produced and their associated diseases. 

## 2. Aerosolization of Moulds Particles from Contamination Materials

Most aerosolization studies focused on moulds as common microorganisms that are capable of growing on building materials, reproducing and then releasing airborne particles into the indoor air under specific conditions and in various forms, such as spores or mycelial fragments, that may contain mycotoxins. The release of fungal particles from colonized sites, e.g., building materials, into the indoor air can be defined as an aerosolization process [16]. In the following section, the mechanisms of aerosolization and different factors influencing the release of particles (fungal spores, fungal fragments) from contaminated surfaces into the indoor air are presented.

### 2.1. The Mechanism of Aerosolization and Characteristics of Airborne Particles

Both passive and active mechanisms may be involved in the release of fungal particles from materials [41]. Active release involves fungal spores and is based on forces arising inside the fungi and is attributable to a burst of energy by osmotic pressure and surface tension discharge [42]. Passive release can involve spores but also other particles and occurs due to energy originating from outside the fungus, such as mechanical disturbances of the colony, e.g., handling, vibrating or ventilating, which may cause release of particles from surfaces [41]. In the indoor environment, human daily activities, such as vacuuming, sweeping, walking, etc., are considered passive mechanisms for the release of fungal particles and have been shown to increase fungal spore concentrations in the indoor air [43]. In addition, the transfer of spores into air strongly depends on their shape and on the organization of conidia in fungal structures (Figure 1). For instance, *Aspergillus* and *Penicillium* spp. are characterized by spores that are organized in long chains (Figure 1a,b), which allow them to be easily released. On the other hand, spores of the *Stachybotrys* spp. are clusters covered with dry slime and are therefore not directly exposed to air flow, which makes it harder for them to become airborne (Figure 1c) [44,45,46]. Thus, the aerosolization processes and mechanisms are affected by the nature of involved micro-organism, some of them being designed to spread more easily in the air (*Aspergillus* spp., *Penicillium* spp., etc.) than others (*Stachybotrys* spp.).

Different studies reported that the release of fungal fragments from contaminated building materials may exceed that of spores. In aerosolization chamber studies, Górny et al. [47] and Cho et al. [48] reported that the release of fungal fragments is 11 to 320 times higher than the release of spores for *Aspergillus versicolor*, 17 to 170 higher for *Cladosporium cladosporioides* and 7 to 270 higher for *Penicillium mellini*, emphasizing that other fungal particles than spores can be aerosolized from a contaminated substrate. By evaluating the size of these particles, researchers were able to distinguish between fungal fragments and fungal spores. In that work, authors based their classification on previous studies that evaluated spore size distributions through the microscopic observations of the studied fungal species. They were 2–3.5 µm for *A. versicolor* (close to spheres), 2–3 µm by 4–7 µm for *C. cladosporioides* (ellipsoidal shape) and 5–6 µm for *P. mellini* (close to spheres) [49]. Based on these data, the particle size of 1.6 µm was selected by Gorny et al. as the lowest size limit separating fungal spores from fungal fragments [47]. They evaluated the air speed necessary to observe the highest ratios of fragments/spore release. The results were 5.8 m/s for *A. versicolor*, 1.4 m/s for *C. cladosporioides* and 0.3 m/s for *P. mellini*, and the highest ratios were obtained at 29 m/s for all species [47]. It has to be noted that air speeds used in that study are very much higher than those normally encountered in buildings. 

After their release, airborne fungal spores and fragments can be considered as solid particles and will behave differently in the air and, if inhaled, in the lungs, based on various particle-related (physical, chemical) and human-related (biological) factors [15]. Physical factors include the morphological characteristics (size, shape, density, electrical charge, etc.) of particles. Chemical factors include composition and hygroscopicity [50]. Regarding physico-chemical factors, it has been reported that the inhalation of airborne particles by human is interdependent on both their size and the period they are actually suspended in the air [51]: microbial particles of 100, 10, 3, 1 and 0.5 μm require 5.8 s, 8.2 min, 1.5 h, 12 h and 41 h to be inhaled, respectively, according to Stoke’s law [52]. Biological host-related factors, such as the breathing pattern, the route of the breath and the anatomy of the airways, will also influence inhalation, as well as fungi-related factors, such as the presence of membranous proteins (e.g., adhesins). Fragments and spores can be both harmful to human health but the fact that fragments are smaller in size than spores provides them a stronger ability to go deeper through the respiratory system when inhaled by humans (Figure 2) [41,47].

For instance, according to their size (5 to 6 µm), *Penicillium mellini* spores may penetrate the respiratory system only to pharynx level, whereas the spores of *Aspergillus versicolor* of 2 to 3.5 µm may penetrate up to bronchi levels. The fragments of different species may penetrate in all the respiratory system as their sizes are <1.6 µm (Figure 2) [54]. These data are obviously to be re-evaluated according to the physiological state of the subject’s bronchi. The effects of airborne particles on human health are discussed in Section 5.

### 2.2. External Factors Impacting the Release of Bioaerosols

There are numerous environmental factors that may impact directly or indirectly the release and spread of bioaerosols in the indoor air. They are related either to the involved microorganisms, to the colonized material, to the occupants’ activities and to the environment that impact both of the growth of microorganisms as well as their aerosolization. 

Microbial-related factors include diversity, age of colonization, spreading and stage of development, etc. As presented previously, certain fungal species are more likely to be aerosolized than others (e.g., *Stachbotrys* spp. vs. *Aspergillus* spp.). It has to be highlighted that there are no data on the impact of multi-species colonization on the release of the different involved species (additivity, synergy or even antagonism towards aerosolization process).

Material factors include hydroscopy, smoothness, roughness, composition, etc. [44,49,55]. Lee et al. [5] studied the release of different fungal particles from different flood-damaged building materials in vitro: linoleum, rugs, carpets and pillows. They observed that, at an air velocity of 0.9 m/s, the total amount of particles released over 10 min was the highest for linoleum (25,503 particles/cm^2^), followed by rugs (1562 particles/cm^2^), carpets (508 particles/cm^2^) and pillows (24 particles/cm^2^), which was correlated with the highest particle concentration observed in linoleum and the lowest observed in the pillow samples. While comparing the duration required for release, Lee et al. detected that time required to aerosolize 90% of the total released particles was shortest for linoleum (<6 s), followed by pillows (<12 s), carpets (24 s) and rugs (78 s), which correlates with the hardness and smoothness of the surfaces [5]. Another study focused on the nature of substrate as an important factor affecting the release of mycotoxins. From different substrates maintained at a constant temperature, relative humidity and luminosity conditions, Moularat and Robine [56] observed clear variations in the release of sterigmatocystin (ST) from *A. versicolor*. The percentage of toxin released compared to the total amount produced on the substrates was 4% in the case of traditional wallpaper, while only 1% of ST was aerosolized from fibre glass and vinyl wallpaper due to the chemical and physical effects of substrate on the aerosolization process [56]. The chemical composition of different substrates provides diverse nutrients and thus different energy levels for the microorganisms to grow. It may therefore directly influence the quantity of toxins produced during fungal development. Considering its further aerosolization, microbial adhesion on the material, microbial development and water retention in the material depend on the structure of the substrate, which would, in this situation, influence the percentage of the toxin to be aerosolized [56].

Environmental factors mainly include turbulence and air velocity, temperature and relative humidity [43,51,57]. The indoor air velocity may be affected by natural ventilation that varies according to the weather and climate and also to the air exchange rates. It can also be influenced by mechanical ventilation due to the presence of ventilators inside the rooms or by heating or air-handling systems. Górny et al. [47] and Aleksic et al. [44] investigated the effect of increasing air velocities on the aerosolization of particles in laboratory conditions. They studied the release of fungal spores and fungal fragments from agar surfaces and ceiling tiles [47] or from wallpapers [44], using specific aerosolization devices. As expected, higher air velocities implied the higher release rates of fungal particles from surfaces into the air in both studies. It was found by Aleksic et al. [44] that air velocities of 0.3 m/s (movement in a room), 2 m/s (mechanical ventilation) and 6 m/s (strong mechanical ventilation or strong draft while opening a window) were able to aerosolize particles of *Penicillium brevicompactum, Aspergillus versicolor* and *Stachybotrys chartarum*, respectively [44]. Moreover, an air velocity of 0.3 m/s was sufficient to release different numbers of particles for *A. versicolor*, *Cladosporium cladosporioides* and *Penicillium mellini* [47].

Relative humidity is also one of the most studied environmental factors that impact aerosolization. Frankel et al. [57] tested the effect of relative humidity (RH) on the release of particulate matters (PM1) (0.54–1.037 µm) and inhalable fractions (1–20 µm) from gypsum boards colonized with *Penicillium* spp. in laboratory conditions. They found that the percentages of gypsum board surface area colonized with fungi correlated positively and significantly with the number of aerosolized particles. In addition, a significantly higher number of PM1 were aerosolized from low RH (22.5 to 27.7%) surfaces (median: 433, range 41–1764 particles/min/cm^2^) compared to those with high RH (94.3 to 96.7%) (median: 83, range 9–303 particles/min/cm^2^). In general, the literature shows that the lower the RH, the higher the concentration of fungal particle in the air [58,59,60]. However, it was demonstrated that particles released under wet conditions have a higher total inflammatory potential (TIP) than those released under dry conditions [57]. It is worth noting that the total inflammatory potential is obtained by using Granulocytes assay (an assay used to assess the microbial contamination of medical drugs) [61]. HL-60 cells are exposed to the particles collected by GSP sampler and react by producing reactive oxygen species (ROS) when exposed to microbial compounds. The total inflammatory potential of a particle correlates positively with the ROS produced [57]. 

Moreover, meteorological factors have also been shown to impact indoor airborne concentrations of bioaerosols. Frankel et al. [62] compared the quantities of airborne particles of fungi and bacteria in the indoor environment according to different seasons. They concluded that: (1) in winter and spring, the main sources of airborne fungi present in the indoor air are originated from indoor environments, while in other seasons, the main sources of fungal airborne particles have outdoor sources. (2) In all seasons, as the outdoor temperature, indoor temperature and AER (air exchange rate) increase, the concentration of indoor airborne particles of fungi increases [62].

In summary, this part of this review highlights that air velocity, material type and relative humidity appear to be, if not the main, the most studied factors in different aerosolization building-related studies. It appears that microbial-related factors are less studied in building environments, thus it is difficult to quantify their impact on particles aerosolization and to compare it with environmental or material factors. In addition, it should be noted that, among the reviewed studies, there is no indication of the relative importance of one of these factors or another on the release of particles. 

These findings provide perspective on the aerosolization of microbial particles from materials, which is a well-known phenomenon in the field of microbiology but surprisingly unstudied in the context of building contamination. This section also emphasizes that the aerosolization mechanisms seems a key parameter in the degradation of IAQ by material-colonizing mould. Having discussed the aerosolization and factors impacting this process in laboratory conditions, the next sections address the contamination of building materials in situ and, later in the paper, the probable relations between building materials contamination and indoor air contamination.

## 3. Microbial Species Identified on Indoor Building Materials

Building-related studies in different European countries and in the USA indicate that more than 600 fungal species may be found in the indoor environment of homes (including those that are flood-damaged, mould-damaged and with no visible mould growth), schools, hospitals, laboratories or care centres [63,64]. Fungi are able to grow in almost all environments if a sufficient amount of water is available. That is why water-damaged buildings present a high susceptibility to fungal growth. Usually, a building is considered as “water-damaged” when building materials are or were subjected to water leakage due to plumbing problems, flooding and/or strong moistening due to water vapor in badly ventilated buildings [9,65]. These buildings are known to be favourable for the growth of various fungal genera [5,66,67]. Epidemiological evidence from studies of indoor air quality and the health of people living in damp or mould-infested buildings suggests that the presence of abnormal amounts of indoor water may increase the risk of respiratory diseases for occupants [68,69].

Many building-related studies investigated the fungal development on surfaces of materials and identified fungal genera that were present [2,70,71]. However, only few studies have gone further and identified fungi at the species level [72,73,74], although such identification is crucial to estimating the potential impact on human health. 

With the aim of listing the most frequently identified fungal species on materials in the indoor environment, Appendix A provides the results of 34 in situ studies from Europe and USA, between years 2001 and 2022, that meet the following requirements: Studies focused on accessible surfaces in buildings;The investigations of fungal development focused on surface sampling;Fungal identification was carried out at the species level.

It should be noted that most of these studies were carried out in mouldy dwellings or dwellings in which inhabitants displayed respiratory problems [9,67,72,73,75,76,77,78,79,80,81,82,83,84,85,86]. Data are less numerous on normal buildings with no dampness, no visible mould contamination and no respiratory problems for occupants [63,66,75,76,81,82,87,88]. Only a few studies were performed in other types of buildings, such as hospitals, archives and museums [64,89,90,91,92,93,94,95]. Along with the type of building and the fungal species detected, Appendix A also presents the sampling and analysis methods carried out. 

Based on the data collected in these studies, over 50 different fungal species were identified from 19 fungal genera on the surfaces of materials. Most of the detected species belong to the genera *Aspergillus, Cladosporium*, *Stachybotrys, Penicillium*, *Alternaria* and *Ulocladium.* Overall, the top ten detected species on the surface of materials from all studies (*n* = 34) are: *Aspergillus versicolor* (detected in 74% of the studies)*, A. niger* (68%), *A. fumigatus* (53%), *Cladosporium sphaerospermum* (41%), *Stachybotrys chartarum* (32%), *Penicillium chrysogenum* (29%), *Aspergillus flavus* (26%), *Alternaria alternata* (21%), *Cladosporium cladosporioides* (18%) and *Ulocladium chartarum* (15%). 

### 3.1. Ecological Characteristics of the Most Detected Species

Among detected species in previous studies, the 10 most frequently detected species are all ubiquitous saprophytic fungi, which mostly reproduce asexually [31,32,96,97,98,99], although some of these species also reproduce sexually, such as *Aspergillus flavus* [28]. The main ecological characteristics of these species are presented in Table 1. 

Most of these species have optimal temperatures for development of about 25–27 °C. However, they can easily develop in a wide range of temperatures, including those commonly found inside buildings. Some species, such as *Cladosporium sphaerospermum* or *C. cladosporioides*, can also grow at low and even negative temperatures, which allows them to develop in cold places, such as fridges, and near windows, where water condensation occurs in winter. Most of the frequently identified species inside buildings are xerophilic and can develop at low water activity (about 0.85), meaning that a mild moistening of materials may be sufficient to reach such values. The main exception is *Stachybotrys chartarum*, which requires water activity of higher than 0.94 to grow. This explains why this species is mostly observed in damp buildings/homes, with its development often resulting from a water damage. It is also important to note that colour is not a way to identify the species, since several grow as black colonies (*Stachybotrys chartarum*, *Cladosporium sphaerospermum*, *Alternaria alternata*) with strong differences in associated risks to health (see Section 5).

### 3.2. Sampling Methods to Analyze the Fungal Contamination of Building Materials

Most studies used swab sampling (25 out of 34 studies) (Appendix A). This is a direct surface sampling method used to collect organic and inorganic contaminants on surfaces. It is known to be more effective on smooth surfaces, such as glass and painted surfaces, than on rough surfaces, such as timber and concrete [118,119]. In most cases, an area of 25 cm^2^ (5 cm × 5 cm square) was swabbed by applying pressure on the surface with the swab and rotating its stick [119,120,121]. Contact plates and scraping methods were clearly less used than swabbing (five and three out of thirty-four studies, respectively), although it has been emphasized that scraping, i.e., bulk sampling, is likely to be more suitable than swabbing when sampling building materials [3]. 

It is important to highlight the lack of data allowing the comparison of recovery efficiency between the different sampling methods carried out on building materials.

### 3.3. Analysis Methods to Investigate Fungal Diversity of Building Materials 

From Appendix A, we can see that twenty-three out of thirty-four studies carried out morphological identification after culture, eight studies carried out DNA analysis and three studies carried out both types of analysis. The strong dominance of cultural methods for identification does not allow a comparison with the species detected using other analysis methods. Even among recent studies, morphological identification after culture is the most widespread analysis method [63,95]. This method is based on the analysis of morphological criteria that are specific to a given fungal genera/species growing on a culture media after incubation in controlled conditions [92,122]. The most frequently used culture media are Czapek Dox Agar (CDA), Yeast Extract Agar (YEA) [94], Malt Extract Agar (MEA) [63,92] and Dicloran Glycerol Agar (DG18) [92]. The medium used may have an influence on the results, as it is favourable for some fungal isolates and non-favourable (or non-optimal) for others; it may therefore influence fungal concentration and diversity. The composition of culture media also plays an important role on fungal morphology [123]. For instance, *Aspergillus niger* is known to form mycelial clumps of average diameter about 1.7 mm, but when adding microparticles, such as talc or aluminium, to the culture media, the average diameter decreased to 0.1 mm, becoming free mycelium [124].

The DNA analysis methods are based on the isolation and determination of DNA sequences that are specific to certain taxonomic groups. Different genes are used for the diagnoses of species such as the fungal internal transcribed spacer (ITS) [67,81,82] and β-*tubulin* gene by using primers Bt2a [86]. The main advantage is that the culture of microorganisms and the subsequent bias resulting from media selection is not always required and the analysis can be performed on freshly collected samples. The drawbacks are as follows: the difficulty of quantifying fungi, which is important for risk evaluation; the concentrations of particles, which is required for DNA analysis to be achieved [125]; and the fact that there is no distinction between viable and dead cells. More details on these methods related to contaminated building materials are given in the review by Verdier et al. [3]. 

Only few studies coupled both cultural and molecular analysis methods, as recommended by the French National Agency for Environment Health (ANSES, Paris, France), to better picture the microflora on surfaces of materials [27]. According to the data presented in Table 1, species detected by cultural methods were also detected by DNA analysis methods and vice versa. Indeed, the interest of coupling these two analysis methods would be to avoid the problem of quantification limits, integrate both living and dead particles (the latter being still allergenic) and limit bias related to the differences in growing speeds between species, etc.

As has been highlighted by researchers and health institutions [3,126] for a long time, there is a clear need for consensus or standardized procedures regarding the investigations of the contamination of materials inside buildings (sampling technique, minimum surface to sample, analysis method, etc.). Such a procedure would be helpful to free oneself from the representation bias inherent in using only a swab for sampling and culture for analysis. Moreover, it would help in conducting reliable and comparable quantitative studies that are lacking but important for health risk assessments related to the aerosolization of fungi from surfaces. 

### 3.4. Species Identification as a Function of Building Type 

Among the top 10 identified species, *Aspergillus versicolor*, *A. niger, A. fumigatus* and *A. flavus* were detected at least once in all type of buildings (Figure 3). *Aspergillus versicolor* and *A. niger* were mostly detected in mouldy dwellings (Figure 3a), whereas *Aspergillus fumigatus* and *A. flavus* were more often found in buildings with no occupant health problems, as well as in hospitals. Previous studies have observed that *A. fumigatus* and *A. flavus* have difficulties absorbing nutrients from building materials [28,29,30,31,127]. Their presence on indoor materials could be more likely related to exchanges with the outdoor environment or with food storage areas than with direct growth on materials [29,32,127]. However, their presence in sensitive premises such as hospitals may be of great sanitary importance due to their role in invasive mycosis. 

*Cladosporium sphaerospermum*, *Stachybotrys chartarum*, *Penicillium chrysogenum*, *Alternaria alternata*, *Cladosporium cladosporioides* and *Ulocladium chartarum* were mostly detected in dwellings with visible fungal development or in which occupants had declared respiratory troubles.

By contrast, they were never reported in hospitals and care centres, which have strict and frequent surface-cleaning protocols (Figure 3e). In other buildings, such as museums and wine cellars, all the top detected species, with the exception of *Ulocladium chartarum*, were found at least once (Figure 3f). 

These findings could be explained by the variation of environmental conditions, implying the presence of good growth conditions for specific species and the absence of appropriate conditions for others, which is in accordance with the known ability of *Aspergillus* spp. to grow on all types of materials and buildings [126] within the minimal availability of favourable conditions.

In addition, detected fungal species at hospitals all belonged to the *Aspergillus* genus because, in fact, examiners have only targeted this fungal genus in regard to its role in human infections (Figure 3e).

### 3.5. Variation of Species Detection between Europe and Northern America 

Table 1 includes twenty-six studies from European countries (France, Poland, Portugal, Denmark, Slovakia, Austria, Greece, Belgium and Germany) and eight studies from Northern America (USA and Canada). Reorganizing the frequency of most detected species as a function of the region demonstrates that the five most detected species were detected both in Europe and in Northern America. Some species that belong to the top six to ten were not detected in any study in the USA or Canada (Table 2). These species are *Aspergillus flavus, Penicillium chrysogenum* and *Cladosporium cladosporioides.* Knowing that the number of studies from Europe are three times higher than those from Northern America and having a common top five in detected species between both continents suggests that the difference in region does not imply a major variation in the species detected. A study on the fungi isolated from different materials in the Blue Mosque and Little Hagia Sophia Mosque in Istanbul showed that these species were also the most frequent in these famous historical buildings located on the borders of Europe and Asia [128].

### 3.6. Quantification of Fungal Particles on Surfaces 

As indicated previously, the quantification of microorganisms or, more generally, of particles, in indoor air bioaerosols provides essential information for assessing the potential health risks for exposed occupants of buildings. In the same way, the quantification of microorganisms growing on the surface of materials would help to correlate or, at least, to easier associate the microbial colonization of materials with indoor air quality. However, the literature shows that there are very few quantitative analyses carried on samples originated from surfaces. Among the thirty-four studies reported in Table 1, only two studies quantified the fungal load on surfaces at the sampling location. Sixt et al. [91] quantified spores in patient’s rooms in hospitals. The average evaluated concentration was 2218 colony-forming units (CFU)/m^2^, while in mouldy museums, the average fungal spore concentration of fungal particles reached values as high as 860,000 CFU/m^2^ [94]. Such a difference is not surprising, considering the possible presence of mouldy objects in museums. At the same time, sampling methods refer only to the use of contact plates in the first study [91] and to contact plates and swabbing in the second [94], which could also influence results. In combination with investigation on bioaerosol in indoor air, these results are helpful to highlight the impact of the surface contamination of materials on indoor air quality degradation. The next section synthesizes studies that carried out in situ sampling on both indoor air and surfaces of materials. 

The analysis of the studies that investigated indoor fungal contamination at the species level shows that some species are of particular importance as frequent contaminants, regardless of country. These species, although common contaminants of many plants, crops or feeds, take advantage of their physiological characteristics to be able to grow in indoor environments. Their presence may have different consequences on the IAQ and health of occupants (see Section 5). However, to date, the lack of reference methods for both sampling and analysing the fungal contamination of materials makes it difficult to draw direct relationships between the presence of these species and the symptoms observed in inhabitants. The few quantitative studies available highlight that the fungal contamination of surfaces can reach quite high values, suggesting a possible direct link with IAQ degradation. The following section will present data demonstrating the link between the mould contamination of materials and indoor air contamination.

## 4. Comparison between Indoor Material and Indoor Air Contaminations

Among the thirty-four building-related studies presented in Table 1, twenty-one investigated the presence of fungi on the surfaces of materials as well as in the air during the same sampling campaign (studies identified by (*) in Appendix A). In these studies, air samples were sampled using MAS (Microbial Air Monitoring System) impactors and analysed following the same analysis methods used for surface samples. Interestingly, the most detected genera from air sampling are *Aspergillus*, *Penicillium*, *Cladosporium* and *Stachybotrys*, which are the same as those detected on surfaces. Moreover, the 10 most detected species in the air were also the most detected on materials. Table 3 presents the frequency of the detection of airborne species and from surface sampling in studies that recorded both surface and air sampling. This shows that the species that are mostly detected airborne are, at the same time, detected on surfaces in the same studies and with approximately an equal frequency of detection (±1). It also demonstrates the frequency of detection of the top 10 most detected species from studies that only carried out surface sampling. 

These data, in accordance with previous studies, indicate that damp indoor environments are favourable sources for airborne *Aspergillus niger* [129], *A. versicolor* [130], *Penicillium chrysogenum* [78] and *Stachybotrys chartarum* [4]. Comparing air and surfaces, we can see that there is a clear association between air and material contaminations for certain species. Nine of the ten most identified species were detected in the same ratio in both environments (Table 3). However, such a correlation does not exist for all species. For instance, *Aspergillus flavus* was detected more often in the air than on the surfaces of materials. In addition, *A. flavus* was less detected in mouldy dwellings and more often in buildings with no health problems, as well as in hospitals (Appendix A). One can assume that this species is not likely to grow on building materials. It may originate from the outdoors and occasionally contaminate building material when exchanges with the outdoors are higher (through air and human activities for example), and thus may be detected only for short periods of time. 

Nevertheless, these findings strongly suggest that materials presenting fungal growth are the main sources of indoor airborne contamination. However, these in situ studies are not sufficient to establish a cause–effect relationship and clarify the mechanisms involved: the contamination of air from surfaces or the opposite. They do not allow the evaluation of the quantitative impact of surface contamination on indoor air degradation. Indeed, for some species that originate from materials but also from other sources (food, human activity, etc.), air sampling cannot differentiate the source of microbial contaminants.

### Quantification of Airborne Contamination by Fungal Particles

The guidelines of the WHO consider that airborne fungal spore concentration (AFSC) above 500 CFU/m^3^ is considered as hazardous for the occupant, and above 1000 CFU/m^3^ is considered extremely hazardous [27]. Figure 4 presents a comparison of airborne fungal spore quantification in normal dwellings, mouldy dwellings and in some hospitals, based on quantitative measurements from 12 different studies.

Among the thirty-four studies presented in Appendix A, eight included the quantification of airborne fungal spores. In mouldy dwellings, AFSC is >1000 CFU/m^3^ in 6 out of 8 studies, which is considered ‘abnormal’ and requires professional intervention for remediation, as recommended by the French Agency, ANSES (Paris, France) [27]. In normal dwellings, with no visible mould and no respiratory health problems, the AFSC is below or around 500 CFU/m^3^. This value was respected in all corresponding studies but one, where the mean level was 535 CFU/m^3^ [76]. This could be due to the fact that, in this study, some of dwellings considered as ‘normal’ were, in fact, contaminated [76]. In the same study, dwellings considered as ‘mouldy’ displayed visible mould growth and the average airborne fungal spores concentration was 1116 CFU/m^3^ [76]. 

Among the studies presented in Appendix A, only those by Sixt et al. [91] and Gutarowska et al. [94] quantified fungal species both on surfaces and in indoor air. In patients’ rooms in a hospital, the average airborne concentration was 33 CFU/m^3^ with an average fungal load on surfaces of 2218 CFU/m^2^. Moreover, in mouldy museums, the average airborne fungal spore concentration reached 3573 CFU/m^3^ with an average concentration of fungal CFU on surfaces of 860,000 per m^2^ [94]. This result is in agreement with the work by Gaüzère et al., who demonstrated that the air contamination of the Louvre with *Aspergillus fumigatus* spores was about 1000 CFU/m^3^ and became stable with time [131].

Although they are very few studies, these data demonstrate that higher concentrations of fungi on surfaces imply higher airborne concentration. At the same time, they also demonstrate that cleaning and disinfection procedures combined with airborne treatment had a significant impact on both levels of contamination. Further studies in this area should investigate the nature and quantification of fungal species that grow on indoor materials, as well as carry out experimentations under controlled conditions to better picture the relation that may exist between surface contamination and subsequent airborne contamination. 

Moreover, among the different reviewed studies on aerosolization, there was no consensus regarding the direct relation between surface and air contamination. In fact, the present review emphasizes this lack of information and the clear need to carry out studies on both indoor elements, namely air and surfaces. This certainly requires a specific methodology to distinguish whether the initial source of airborne particles is surface contamination or another type (such as outdoor air, for instance).

## 5. Possible Consequences on Human Health of Fungal Particles Release in Indoor Air

As mentioned previously, the fungal species that are commonly detected indoors may represent a risk for human health. Among the top ten species found in indoor environments, most are either directly pathogenic (responsible for fungal infections) or indirectly toxic due, as an example, to the production of allergenic or toxic (mycotoxins) compounds. Table 4 summarizes the mycotoxins produced by the most commonly detected species as well as their associated diseases.

*Aspergillus versicolor* was reported to be a causative agent of aspergillosis and a major cause of onychomycosis (fungal infection of the nails) [149]. Mycotoxins produced by this species act as immunosuppressants, resulting in the increased prevalence of infections among inhabitants of damp buildings [134]. In a case study, it was shown that *A. versicolor* was also a causative agent of invasive aspergillosis [150]. *A. versicolor* is known as a major producer of the hepatotoxic and carcinogenic mycotoxin sterigmatocystin [130,133].

Interestingly, *Aspergillus niger* has been reported to be less likely a cause of human disease compared to other *Aspergillus* species [149], although immune-compromised individuals are susceptible to *A. niger* infections, causing, in most cases, invasive pulmonary aspergillosis [151]. *A. niger* is capable of producing carcinogenic mycotoxins, such as ochratoxin A [135].

*Aspergillus fumigatus* was associated with a range of pulmonary infections [152], and it has been reported to produce metabolites, such as allergenic polypeptides, that are responsible for asthma and rhinitis [33]. *A. fumigatus* is known to produce various immune-suppressive mycotoxins, such as gliotoxin, that have negative effects on human and animal health, while fumagillin, helvolic acid, fumitremorgin A and asphemolysin [153] as well as β-1,3 glucans are known modulators of the immune system [154,155].

*Cladosporium* spp. are not major producers of mycotoxins [156,157] but cause allergies and produce volatile organic compounds (mVOCs) associated with odours [157]. For instance, *Cladosporium sphaerospermum* was reported to cause allergies and other occasional diseases in humans [157]. However, several studies have reported the production of toxic compounds by *C. sphaerospermum* and *C. cladosporioides* species. Some metabolites produced by these species have antifungal and antibacterial activities, such as deacetyl-yanuthone A, 1-hydroxyyanuthone A and ophiobolin K [146], while others are toxic for eukaryotic cells, such as atenuisol, calphostin A, C, D, cladosporol and emodin, which are inhibitors of protein kinase C [146]. 

*Stachybotrys chartarum* species produce atranone mycotoxins that induce pulmonary inflammation [158]. The fungus also produces macrocyclic trichothecenes, which are very potent protein synthesis inhibitors, and stress kinase activators that appear to be a critical underlying cause for a number of adverse effects [158,159]. This species was associated with idiopathic pulmonary haemorrhages in children in the USA and is suspected to play a direct role in the sick building syndrome [160].

*Penicillium chrysogenum* is a less common pathogenic fungi belonging to the genus *Penicillium* [129]. It is a causative agent of necrotizing esophagitis and asthma [161]. It produces a nephrotoxic mycotoxin, named citrinin [142], and roquefortine C in water damaged building materials [143]. 

*Aspergillus flavus* was associated with clinical syndromes, such as chronic granulomatous sinusitis, keratitis, cutaneous aspergillosis, wound infections and osteomyelitis [30]. *A. flavus* is one of the two main producers of the most important mycotoxin for public health worldwide: aflatoxin B1 (AFB1), which is carcinogenic for humans (group I of the IARC) [142]. This species is also a producer of sterigmatocystin, cyclopiazonic acid, kojic acid and aflatrem [111]. However, the deleterious effects of AFB1 are more associated with food consumption (spices, maize, peanuts) than indoor exposure. Nevertheless, the inhalation of AFB1 was demonstrated to be very toxic and could lead to toxic effects in case of occupational exposure (exposure of workers in cereal silos) [30,162]. As said previously, the presence of *A. flavus* indoors is often related to outdoor air exchange or organic sources of contamination.

*Alternaria alternata* is well known as a producer of various mycotoxins, including alternariol (AOH); alternariol monomethyl ether (AME); altenuene (AE); altertoxins I, II and III; tenuazonic acid (TeA); and tentoxin (TEN) [163,164,165,166,167,168]. AOH and AME were reported as mutagenic and genotoxic [168]. They inhibit cell proliferation and are cytotoxic to HCT116 human cells [166,168]. 

*Ulocladium* species are responsible for food spoilage. They are also plant pathogens. However, they are not important producers of mycotoxins [148]. For instance, *U. atrum* is the only species of this genus able to produce curvularins, while other species may produce infectopyrones and derivatives of altertoxin I [147]. Regarding *U. chartarum*, it has been reported that this species belongs to the same family of *Alternaria alternata* [147,169]. The species is reported as allergenic [169] but produces none of the *A. alternata* toxins [147].

These data highlight the possible toxicity of each of the fungal species belonging to the top 10 of the detected species on surfaces of materials in indoor environments and indoor air. 

As a practical illustration, in some studies, presented in Table 1, investigations on the association between fungal concentration and respiratory health problems were achieved. For instance, Reboux et al. [72] compared results obtained from 908 patients’ homes with 104 habitations in which no health troubles were recorded. In this investigation, observed clinical signs were conjunctivitis, rhinitis, asthma and rhinitis-associated asthma. The authors found that the mean fungal air concentrations in patients’ dwellings were four times higher than those of dwellings in which no health troubles were noted (285 vs. 89 CFU/m^3^, respectively). Moreover, it was demonstrated that asthma signs correlate positively with *Aspergillus versicolor*, *A. niger* and *Alternaria alternata* presence [72]. Similar findings had been reported previously by Reboux et al. [76] and Bellanger et al. [75], who demonstrated a positive correlation between allergic symptoms in inhabitants and the presence of *Aspergillus versicolor, Penicillium chrysogenum* and *Stachybotrys chartarum* in homes.

The direct role of mycotoxins in symptoms occurring in the inhabitants of mouldy homes has yet to be determined. Since involved mycotoxins are often different from those contaminating food, toxicological data are often missing to interpret the possible consequences of the presence of those toxic compounds in indoor environments. Moreover, inhalation may lead to a different behaviour of toxins in the organisms, for instance, avoiding liver metabolism that occurs after ingestion.

It was demonstrated, in laboratory conditions, that certain mycotoxins, such as sterigmatocystin or satratoxins, could be aerosolized from mouldy materials and that they could be borne by inhalable particles [44]. Some studies also demonstrated the presence of certain mycotoxins in indoor air (Table 5). The toxicity of detected compounds at measured concentrations has still to be elucidated. Moreover, the presence of different mycotoxins may also raise the question of the impact of such a multi-contamination, as for foods [170].

The specific effects associated with each of these mycotoxins highlights the importance of identifying fungal contaminants at the species level and not only at the genus level in order to evaluate the risk associated with the fungal contamination of buildings. Moreover, quantitative studies on fungal species from both surfaces and the air, as well as mycotoxin aerosolization and toxicity after inhalation, are required to obtain evidence of causal relations between species developing indoors and occupants’ health. To date, no study has examined the cumulative effect of the presence of several fungal species.

## 6. Conclusions

The main goal of this review was to examine the available literature on the fungal contamination of materials in buildings and in indoor air in order to better understand (i) the connections that exist between the growth on materials and the deterioration of IAQ by bio-aerosols and (ii) the subsequent sanitary risks that could result from fungal development and particles aerosolization. In addition, this review is of significance regarding public health, as it gathers a list of microbial species that are commonly detected on indoor surfaces and in the air, along with the harmful mycotoxins they are susceptible to releasing following aerosolization from materials, and which could be inhaled by the occupants of mouldy buildings. The processes of the aerosolization of fungal particles may be impacted by many parameters, including environmental (air flux, humidity, temperature, etc.), microbial (strains, stage of growth, etc.) and material (surface roughness, etc.). Today, climate change, together with changes in construction methods, may strongly influence these main parameters and thus influence aerosolization. Increased temperatures, air conditioning and bio-based materials used in construction, as well as low-consumption buildings that are very well insulated but may be subject to mould development in cases of poor ventilation, can promote the development of certain heat-preferring and hydrophilic species that are already present and perhaps allow the appearance of new species that are yet to be identified. Continuous monitoring is needed to anticipate any possible developments.

## Figures and Tables

**Figure 1 toxins-15-00175-f001:**
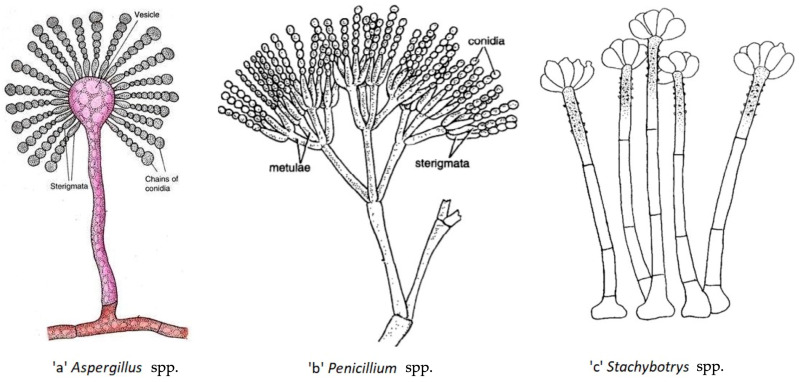
Mycelium and conidiophore organization in *Aspergillus* spp. (**a**), *Penicillium* spp. (**b**) and *Stachybotrys* spp. (**c**); 1a and 1b: long chains of spores easily dispersed by a sufficient air flow, 1c: conidia grouped as cluster and covered by a slime [46,47,48].

**Figure 2 toxins-15-00175-f002:**
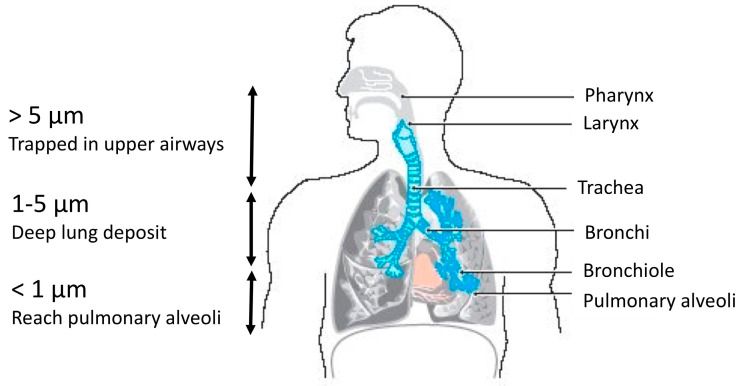
Relation between the size of particulate matters (PM) and their ability to penetrate the respiratory tract of human (adapted from Costa et al. [53]).

**Figure 3 toxins-15-00175-f003:**
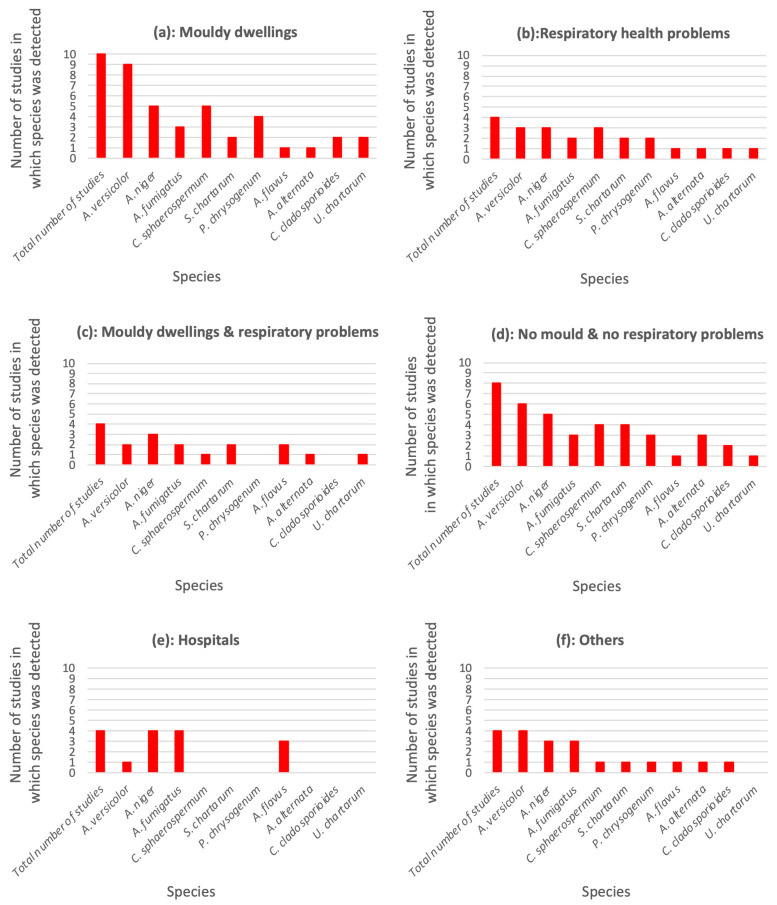
Frequency of detection of the most identified species in: mouldy dwellings (**a**); dwellings with occupants suffering from respiratory problems (**b**); mouldy dwellings in which occupants display respiratory problems (**c**); dwellings with no visible mould and no identified occupant respiratory problems (**d**); hospitals (**e**); others (**f**).

**Figure 4 toxins-15-00175-f004:**
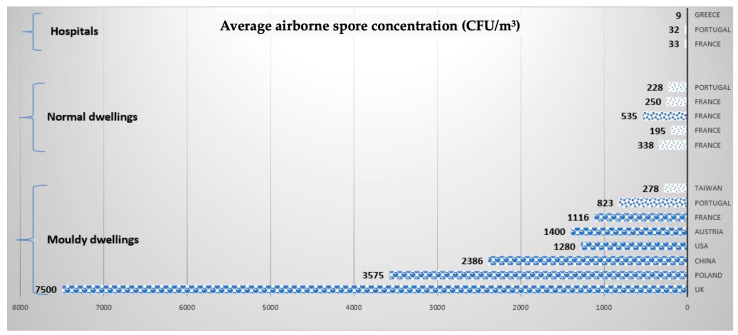
Airborne fungal spore concentrations in different types of buildings from different countries [64,76,84,85,90,91,92,94,130,131,132,133].

**Table 1 toxins-15-00175-t001:** Main physiological characteristics of the top 10 fungal species detected in dwellings.

Species	Temperature for Development		Minimal Water Activity	Colour of the Colony	References
	Optimal	Range			
*Aspergillus versicolor*	22–26 °C	4–40 °C	0.75–0.81	Variable according to environmental conditions: white, pinkish, from yellow to green	[100,101,102]
*Aspergillus niger*	27 °C	25–40 °C	0.85	Turns to black during sporulation	[96,103]
*Aspergillus fumigatus*	37 °C	Up to 55 °C Survive at 70 °C	>0.9	Blue-green	[31,32,104]
*Cladosporium sphaerospermum*	25 °C	−5–35 °C	>0.82	Black	[105,106]
*Stachybotrys chartarum*	23–27 °C	2–40 °C	>0.94	Black	[107,108]
*Penicillium chrysogenum*	20–30 °C	5–37 °C	>0.78	Blue-green	[109,110]
*Aspergillus flavus*	32–36 °C	10–50 °C	>0.78	Bright yellow-green	[99,104,111]
*Alternaria alternata*	25–29 °C	2–32 °C	>0.85	Black	[97,112,113,114]
*Cladosporium cladosporioides*	18–28 °C	−10–35°C	>0.88	Black	[36,97,115]
*Ulocladium chartarum*	25 °C	5–34 °C	>0.9	Golden brown to blackish brown	[97,116,117]

**Table 2 toxins-15-00175-t002:** Most detected species in Europe and in Northern America.

Species	Any location [*n* = 34; (%)]	Europe [*n* = 26; (%)]	Northern America [*n* = 8; (%)]
*Aspergillus versicolor*	25 (74%)	20 (77%)	5 (63%)
*Aspergillus niger*	23 (68%)	17 (65%)	6 (75%)
*Aspergillus fumigatus*	18 (53%)	17 (65%)	1 (13%)
*Cladosporium sphaerospermum*	14 (41%)	12 (46%)	2 (25%)
*Stachybotrys chartarum*	11 (3%)	7 (27%)	4 (50%)
*Penicillium chrysogenum*	10 (29%)	10 (38%)	0 (0%)
*Aspergillus flavus*	9 (26%)	9 (35%)	0 (0%)
*Alternaria alternata*	7 (21%)	6 (23%)	1 (13%)
*Cladosporium cladosporioides*	6 (18%)	6 (23%)	0 (0%)
*Ulocladium chartarum*	5 (15%)	4 (15%)	1 (13%)

*n*: number of studies; % = percentage of studies in which the species were identified.

**Table 3 toxins-15-00175-t003:** Frequency of detection of the top 10 most detected species from all studies that performed surface sampling and from studies that carried out both surface and air sampling.

Species	Studies That Carried Out Both Air and Surface Sampling		Studies That Carried Out Only Surface Sampling
	Number (%) of Airborne Detections [*n* = 21]	Number (%) of Surface Detections [*n* = 21]	Number (%) of Surface Detections [*n* = 34]
*Aspergillus versicolor*	18 (86%)	18 (86%)	25 (74%)
*Aspergillus niger*	15 (71%)	14 (67%)	23 (68%)
*Aspergillus fumigatus*	9 (43%)	9 (43%)	18 (53%)
*Cladosporium sphaerospermum*	9 (43%)	9 (43%)	14 (41%)
*Stachybotrys chartarum*	6 (29%)	5 (24%)	11 (32%)
*Penicillium chrysogenum*	7 (33%)	7 (33%)	10 (29%)
*Aspergillus flavus*	7 (33%)	5 (24%)	9 (26%)
*Alternaria alternata*	4 (19%)	5 (24%)	7 (21%)
*Cladosporium cladosporioides*	5 (24%)	4 (19%)	6 (18%)
*Ulocladium chartarum*	3 (14%)	4 (19%)	5 (15%)

*n*: number of studies.

**Table 4 toxins-15-00175-t004:** Main mycotoxins produced by fungal species most frequently detected on surfaces of materials and the associated diseases in humans and/or animals.

Species	Mycotoxins Produced	Associated Diseases	References
*Aspergillus versicolor*	Sterigmatocystin, nidulotoxins, Aspergillomarasmine A, Aspergillomarasmine B, xanthones, fellutamides and anthraquinones	Allergic diseases, aspergilloses, onychomycosis, immunosuppression	[129,132,133,134]
*Aspergillus niger*	Ochratoxin A (OTA), fumonisin B2 (FB2), fumonisin B4 (FB4)	Aspergillosis, otomycosis kidney failure (OTA)	[135,136]
*Aspergillus fumigatus*	Gliotoxin, fumagillin, fumigaclavines, helvolic acid, fumitremorgin A and Asphemolysin, β-1,3 glucans	Pulmonary infections, modulation of the immune system, abortions in farm animals	[62,63,64,65]
*Cladosporium sphaerospermum*	Atenuisol, calphostin A, C, D, cladosporol, emodin	Inhibition of protein kinase C	[76]
*Stachybotrys chartarum*	Atranones, macrocyclic trichothecenes	Pulmonary inflammation, Protein synthesis inhibition	[66,67,68]
*Penicillium chrysogenum*	Citrinin, roquefortine C, meleagrin, chrysogine, fungisporin, andrastin A	Necrotizing esophagitis, and asthma	[137,138,139,140,141,142,143]
*Aspergillus flavus*	Aflatoxins, sterigmatocystin, cyclopiazonic acid, kojic acid and aflatrem	Hepatocarcinoma (aflatoxins = group I of the IARC), immunosuppression, invasive aspergillosis, allergies	[30,111,139,144]
*Alternaria alternata*	Alternariol (AOH), Altenuene (AE), Alternariol mono-methyl ether (AME), tentoxin (TEN) and Tenuazonic acid (TeA)	Cytotoxicity for animal cells. Fetotoxic and teratogenic to mice and hamsters.	[65,66,67,68,69]
*Cladosporium cladosporioides*	Atenuisol, calphostin A, C, D, cladosporol, emodin,	Allergic reactions, Inhibition of protein kinase C	[145,146]
*Ulocladium chartarum*	Infectopyrones and derivatives of altertoxin I	Allergic diseases	[147,148]

**Table 5 toxins-15-00175-t005:** Mycotoxins identified in indoor air samples in the USA and Europe.

Country	Number of Locations	Detected Toxins	Toxin Concentration	References
USA	7 water-damaged homes with *Stachybotrys* contamination: 40 air samples from 16 rooms	Macrocyclic trichothecenes	<10 to >1300 pg/m^3^ of sampled air	[171]
Belgium	7 water-damaged homes: 20 air samples	Roquefortine C (in 1 sample) Chaetoglobosin A (in 3 samples) Sterigmatocystin (in 3 samples) Aflatoxin B1 (in 5 samples) Aflatoxin B2 (in 4 samples) Roridin E (in 3 samples) Ochratoxin A (in 3 samples)	Roquefortine C: 1–4 ng/m^3^ Chaetoglobosin A: 0.0067–3.4 ng/m^3^ Sterigmatocystin: 0.0034–1.77 ng/m^3^ Aflatoxin B1: 0.0024–0.15 ng/m^3^ Aflatoxin B2: 0.0003–0.02 ng/m^3^ Roridin E: 0.0031–0.08 ng/m^3^ Ochratoxin A: 0.01–0.23 ng/m^3^	[172]
France	15 flooded dwellings with *Stachybotrys* contamination + 9 control dwellings	Macrocyclic trichothecenes	0.29–0.62 ng/m^3^	[173]
Germany	1 air sample form water-damaged building	Satratoxin G and H	Satratoxin G: 0.25 ng/m^3^ Satratoxin H: 0.43 ng/m^3^	[174]
Sweden	37 samples from 22 water-damaged building	Trichodermol (in 26% of samples) Verrucarol (in 21.6%) Sterigmatocystin (in 0.5%) Gliotoxin (in 2.7%) Satratoxin G (in 5.4%) Satratoxin H (in 5.4%) Aflatoxin B1 (in 5.4%)		[175]

## Data Availability

Data presented in this study are available on request to the corresponding author.

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
