# Peer review of "Fungal Contamination of Building Materials and the Aerosolization of Particles and Toxins in Indoor Air and Their Associated Risks to Health: A Review"

_toxins, 2023, doi:10.3390/toxins15030175_

Round 1
Reviewer 1 Report
The review will be a valuable contribution to an important field once it is modified on the lines suggested below.
1. There are minor errors ibn he English throughout the manuscript which need to be corrected before it is in a publishable form - see comments on the abstract for examples where text can be improved.
Ln 12insert "the" before "occupants health"
Ln 16 delete "works" insert "studies"
Ln 17 delete "mouldy" insert "in buildings where mould was a contaminant"
Ln 18 delete "and" insert "and there was" a strong...
Ln 20 location insert "in" Europe "or the" USA
Ln 22 delete capital letter "H" should be "h"
Ln23 should be "work is"
Ln-24-27 rewrite as 2 sentences
and so on in the rest of the manuscript
Ln 49 Due to their .. delete "reliability" substitute "reliance"
Ln 95 "AND" should be lower case
Ln 125 delete "a mechanism known as"
Ln 20 to 239 the data discussed would be better presented as a Table with comments in the text
Ln 280 Delete "capable" insert " "able"
Ln 305 Table 1 if retained in its current form this Table should be in an Appendix and the key information summarised in a smaller Table? See LN 309 to 316.
Could Ln 322 to Ln 357 be put in a Table and summarised in the text?
Figure 3 The type face used for the axes is too small - modify so is clearly presented for the average reader
Reviewer 2 Report
General comments:
I have reviewed the review entitled “Fungal contamination of building materials, aerosolization of particles and toxins in the indoor air and associated risk for health: a review” (toxins-2218741).
After my review of the report, it is my judgment that it is a systematic and comprehensive article. However, necessary corrections should be made before publication.
All the marked places in the PDF version must be revised.
There are some examples of my specific comments as follow.
1. I strongly suggest the authors to read the ‘Intructions for Authors’ of Toxins carefully, the uniform format should be used throughout the text, such as the citation format of references: [5–8] in Line 44, compounds [13-15] in Line59, Górny [47] in Line 146, Górny et al [47] in Line 157, etc.
2. I suggest the authors to double check all the references listed in the “References” section.
3. The citation of references in section ‘1. Introduction: importance of moulds as indoor contaminants’ is mixed or flawed. For example, references 21, 28, 35, 38, and 39 were missed in the main text.
4. The use of ‘spp’ in the whole text is wrong, ‘spp.’ should be used and it doesn’t need italics.
5. There should be a space between the number and the unit. For example, in Lines 186, 189, 235, 322–356, etc.
6. I suggest the authors to re-prepare the Table 1, the current version is difficult to follow.
7. I suggest the authors to provide some examples about fungal contamination of some worldwide famous buildings, such as the Louvre, the Sagrada Familia Cathedral, the Palace Museum, and so on.
8. What’s the significance of the review? It’s better to emphasize the main contribution of this review in the manuscript.

Round 2
Reviewer 1 Report
The layout and text of the manuscript is much improved. There seems to be some confusion with the labelling of figure 4 (Ln 515) -the original caption is deleted. On Ln 612 there is a new Figure 4 with caption?
Author Response
Dear Reviewer,
once again the authors thank you for your appreciation of our work.
Regarding your remark on a new figure 4, it seems that the problem may be linked to the "track changes version" of the manuscript in pdf that was quite hard to follow and indeed let think that there was a new figure (in fact, a previous version of the figure 3 removed from the revised version).
We hope that in the new revised version, it will be clearer.
Reviewer 2 Report
I appreciate the authors for their responses to my comments. The authors have improved the manuscript, but there are still some obvious mistakes in the main text and my specific comments as follow.
Specific comments:
1. Lines 273 and 701, there should be a space between …) and [..].
2. Lines 344 and 349, ‘and’ should not be italic.
3. Lines 346, (n=34) should be (n = 34).
4. Line 411, the use of ‘x’ in ‘5 cm x 5 cm square’ is not correct.
5. Lines 437, gene should be italic.
6. Some references should be corrected in section References marked with yellow.
